# RecombiCraft library construction: A novel method for DNA library cloning and expansion using non-enzymatic single-step DNA recombination and liquid culture

Yuki Kawai-Harada[1,2], Mehrsa Mardikoraem[1,3], Katherine Lauro[1,2], Vasudha Nimmagadda[1,4], Quynh Tong[1,4], Kayla Bello[1,4], Daniel Woldring[1,3], Masako Harada[1,2]*

1 Institute for Quantitative Health Science and Engineering, Division of Chemical Biology, Michigan State University, East Lansing, Michigan, United States of America, 2 Department of Biomedical Engineering, College of Engineering, Michigan State University, East Lansing, Michigan, United States of America, 3 Department of Chemical Engineering and Materials Science, Michigan State University, East Lansing, Michigan, United States of America, 4 Lyman Briggs College, Michigan State University, Michigan, East Lansing, Michigan, United States of America

* mashar@msu.edu

**Data Availability Statement:** The data that support the findings of this study are openly available in

## Abstract

In this study, we introduce **RecombiCraft**, an innovative, rapid, and cost-efficient method for constructing DNA libraries in *E. coli*. This method uses seamless ligation cloning extract (SLiCE) coupled with liquid culture amplification to effectively minimize sequence biases. The technique capitalizes on the natural homologous recombination capabilities of *E. coli* cell lysates, eliminating the need for multiple purified enzymes and reducing costs. We first synthesized the library backbone and inserts via PCR, employing high-fidelity polymerase to minimize sequence bias. The SLiCE technique was then used to assemble the DNA fragments introduced into *E. coli* through electroporation. To ensure the integrity of the library, we optimized culture times based on next-generation sequencing analysis which confirmed the minimal sequence bias. The RecombiCraft method demonstrates that this approach is economical and maintains the library's uniformity. By using liquid culture, this method can complete DNA library generation in about 12 hours and final extraction is simple, making it a promising tool for genetic research and biotechnology applications.

## Introduction

DNA libraries play a pivotal role in modern biological research by providing comprehensive collections of genetic information essential for understanding the complexities of life. These libraries serve as tools for elucidating gene function, regulatory mechanisms, and disease pathways, thereby facilitating the discovery of new drugs and biotechnological innovations [1].

The diversification of DNA libraries is achieved through natural, synthetic, or semi-synthetic methods. Traditional natural libraries are composed of genetic material from a variety of sources, including humans and other mammals, selected based on specific attributes such as

Figshare at http://doi.org/10.6084/m9.figshare.27183669.

**Funding:** This project received financial support from two sources: the Elsa U. Pardee Foundation, which focuses on cancer research and treatment funding, and the Michigan Translational Research and Commercialization (MTRAC) program, which provides early-stage funding to promote the commercialization of innovative technologies.

**Competing interests:** The authors have declared that no competing interests exist.

age, phenotype, or immunization status for antibody libraries [2]. Synthetic libraries, on the other hand, are derived from pre-existing genetic templates modified through either random or directed mutations. Techniques like error-prone PCR intentionally reduce polymerase accuracy to induce mutations broadly across the genome [3, 4], while strategies such as chain shuffling rearrange the sequence order [5, 6]. Targeted approaches involve modifications of specific sequences or regions, often utilizing degenerate oligonucleotides for targeted changes [7–9]. Nicking mutagenesis is a modern technique that introduces specific mutations into DNA by creating nicks in one strand and using them as starting points for targeted changes during replication or repair [10].

The integration of DNA fragments from these libraries into a backbone is generally categorized into two main strategies: 1) those employing restriction enzyme sites, and 2) those leveraging the homology between the insert and backbone. Common examples of the first category include techniques like Golden Gate Assembly and blunt-end cloning [11–15]. The second category encompasses methods such as ligation-independent cloning (LIC), Gibson assembly, and seamless ligation cloning extract (SLiCE) [16–20]. While both categories are widely used in constructing DNA libraries, LIC or Gibson Assembly rely on multi-enzyme reactions, which can be relatively costly due to the need for multiple purified enzymes. In contrast, as previously stated, SLiCE utilizes the natural homologous recombination capabilities of *E. coli* cell lysates, offering a more economical alternative by using readily available laboratory strains. However, despite these advantages, there have been no reports of SLiCE being applied to the construction of DNA libraries.

In this study, we present RecombiCraft, a novel and budget-friendly method to construct DNA libraries in *E. coli* by SLiCE, followed by a liquid culture that amplifies the libraries and ensure minimal sequence bias.

## Material and methods

### Backbone and library insert synthesis

The library backbone was created by PCR amplification. The reaction contains 300 μM dNTP, 300 nM each of forward/reverse primer (S1 Table), 0.5 U KAPA HiFi DNA polymerase (07958889001, Roche Diagnostics), 1x KAPA HiFi buffer, and 1 ng template library DNA [21] in a total reaction volume of 20 μL using T100 Thermal Cycler (BIO-RAD). The PCR amplification cycle was as follows: Backbone synthesis 95°C for 3 min; 30 cycles of 98°C for 20 sec, 65°C for 15 sec, 72°C for 2 min; 72°C for 1 min for final extension, Insert synthesis 95°C for 3 min; 30 cycles of 98°C for 20 sec, 65°C for 15 sec, 72°C for 30 sec; 72°C for 1 min for final extension. PCR product was treated with 40U Dpn I for 1 hour and was purified by QIAquick PCR Purification Kit (QIAGEN). The purified DNA was quantified by Qubit dsDNA BR kit (Invitrogen).

### SLiCE reaction

The purified backbone and library inset were assembled by SLiCE. The reaction contains 1x SLiCE buffer, 1uL SLiCE extract, and 100 ng of DNA (backbone-to-insert ratio was 1:10) in a total reaction volume of 10 uL. The reaction mixture was incubated at 37°C for 1 hour followed by purification using QIAquick PCR Purification Kit (QIAGEN). The purified DNA was quantified by Qubit dsDNA BR kit (Invitrogen).

### Electroporation

Assembled library DNA was introduced to NEB® 10-beta Competent E. coli (C3020K, NEB) by using MicroPulser Electroporator (BIORAD). Before electroporation, 50 uL electro-

competent cells were aliquoted in a microcentrifuge tube and placed on ice for 10 min, and then 100 ng of SLiCE product was added. Competent cells were transferred to 1mm electroporation cuvette (ThermoFisher), and electroporation was done with a preinstalled EC1 program by following the manufacturer's protocol followed by preincubation at 37˚C for 1 hour in 950 uL Stable Outgrowth Medium.

### Bacteria culture and library DNA extraction

Preincubated cells were cultured at 37˚C for the scheduled time in 50 mL LB media with 100 ug/mL Ampicillin. Bacteria were harvested by centrifuge, and library DNA was extracted by ZymoPURE II Plasmid Midiprep kit (Zymo research). For CFU measurement and control sample preparation for next-generation sequence analysis, preincubated cells were plated on LB agar plate with 100 ug/mL, and incubated at 37˚C overnight.

### Next-generation sequencing

The sample for next-generation sequencing (Illumina MiSeq) were prepared by PCR amplification using a primer pair (S1 Table) with a sequencing index for the sequencing library amplification. A fluorometric method (Qubit) was used to quantify the dsDNA PCR products before the submission. All the samples were normalized to the same concentration and agarose gel electrophoresis was used to confirm the product size. Sequencing was performed at the MSU Genomics Core facility using MiSeq Reagent Kit v3 for 250 bp paired-end (PE) reads. The generated FASTQ format file was extracted, processed and clustered by sequence similarity using our custom software, ScaffoldSeq5.

## Result

### SLiCE-based DNA library construction

The SLiCE reaction is a cost-effective and efficient cloning method using bacterial lysates [19, 22]. By designing homologous sequences at both ends of the backbone and insert, it is possible to obtain the desired plasmid through homologous recombination. Fig 1 shows a schematic diagram of library DNA construction applying this technology. Briefly: **1)** Homologous Sequences: The backbone and library insert have homologous sequences at both ends; **2)** Circularization: These sequences are circularized using the SLiCE reaction; **3)** Introduction into

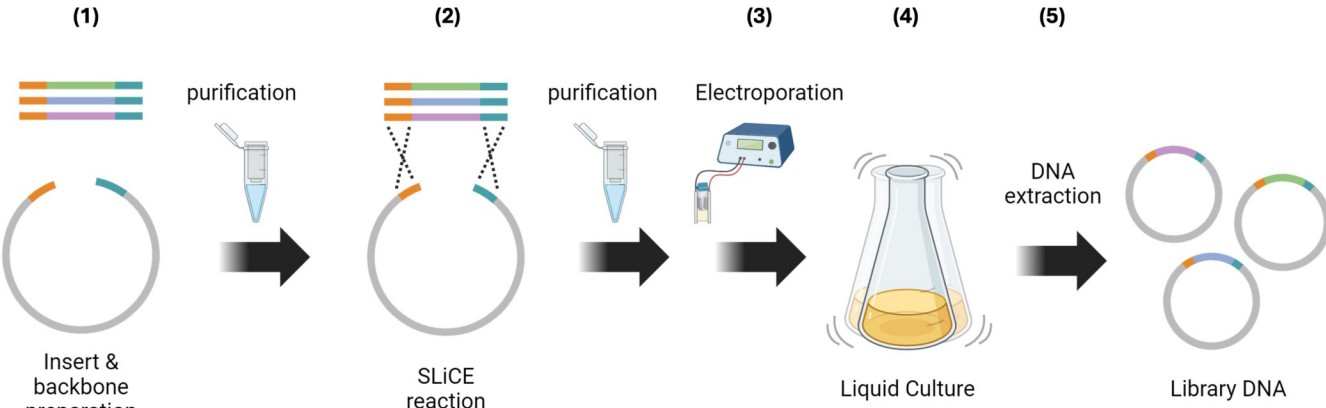

**Fig 1. Schematic diagram of library DNA construction.** Vector and library insert are prepared by PCR followed by purification. The library assembled by SLiCE is introduced to *E.coli* by electroporation and amplified through liquid culture. This image was created with BioRender.com.

E. coli: The circularized DNA is introduced into E. coli cells via electroporation; **4)** Growth in Liquid Culture: The E. coli cells containing the library DNA are grown in liquid culture after preincubation; **5)** library DNA Extraction: Finally, the amplified library DNA can be extracted using a standard plasmid DNA extraction method.

## Optimization and characterization of SLiCE library construction

It has been shown that the efficiency of SLiCE cloning depends on the length of the terminal homologous sequences. We improved library cloning by creating different-length homologous sequences and comparing their cloning efficiency. To minimize PCR bias during backbone and library insert amplification, KAPA HiFi DNA Polymerase was used for PCR amplification [23]. A combination of approximately 50 bases (57 bp on the 5' side and 52 bp on the 3' side) showed the highest efficiency (Fig 2). This trend aligns with the report by Zhang et al [19]. While liquid culture of E. coli is desirable for efficient DNA extraction, concerns arise about differences in clone growth rates affecting library uniformity [24, 25]. To address this, we investigated bias due to liquid culture using next-generation sequencing (NGS). We prepared 20 different monobody sequences with variable binding loop regions [21]. Monobodies are synthetic binding proteins consisting of fibronectin type III domain (FN3) [26, 27]. After introducing a plasmid DNA solution containing equal numbers of molecules of 20 monobody sequences into electrocompetent cells, we cultured them in an antibiotic-supplemented LB medium. Library DNA was extracted at various time points (2H, 4H, 8H, 12H, and 16H) and analyzed by NGS. Comparing cells cultured on LB agar plate as a control, we observed no significant variance change from 2 to 8 hours. However, a significant increase in variance occurred after 12 hours (Fig 2B). Based on these results, we decided to set the culture time for library construction to 8 hours.

**Assessment of library DNA generated by optimized construction method.** The monobody library was designed based on a hydrophilic variant of human fibronectin, which improves biodistribution properties, making it well-suited for downstream applications in molecular imaging and therapeutics. This hydrophilic monobody library included diverse

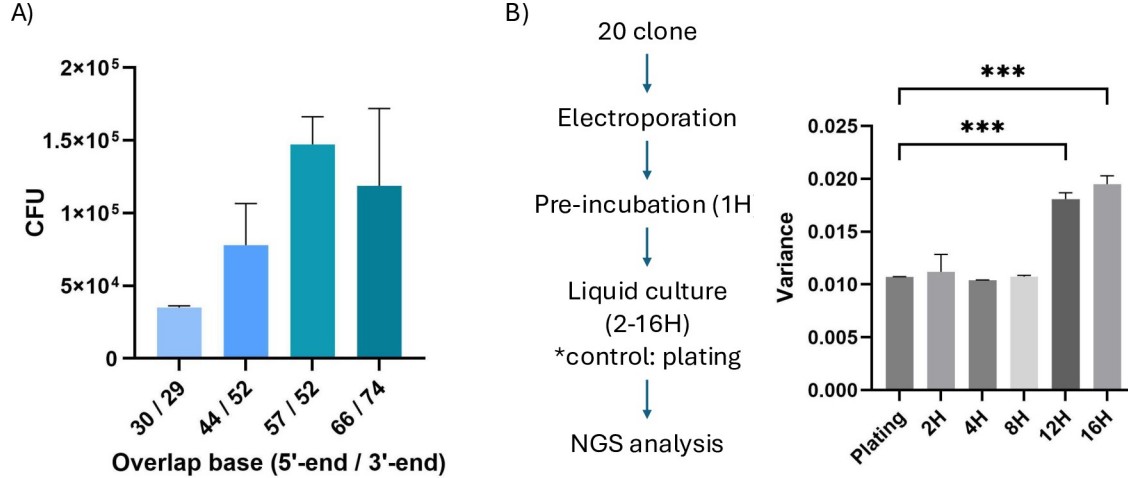

**Fig 2. Optimization and verification of DNA library construction.** (A) Optimization of the sequence length for homologous recombination. An overlap of approximately 30 to 70 bases was designed for homologous recombination, and the efficiency of the SLiCE reaction was compared (n = 2). (B) Evaluation of bacteria growth bias due to liquid culture time. DNA uniformly containing 20 types of plasmids randomly selected from the Monobody library was introduced into E. coli by electroporation, and the growth bias of each plasmid-retaining strain at each culture time was analyzed by NGS.

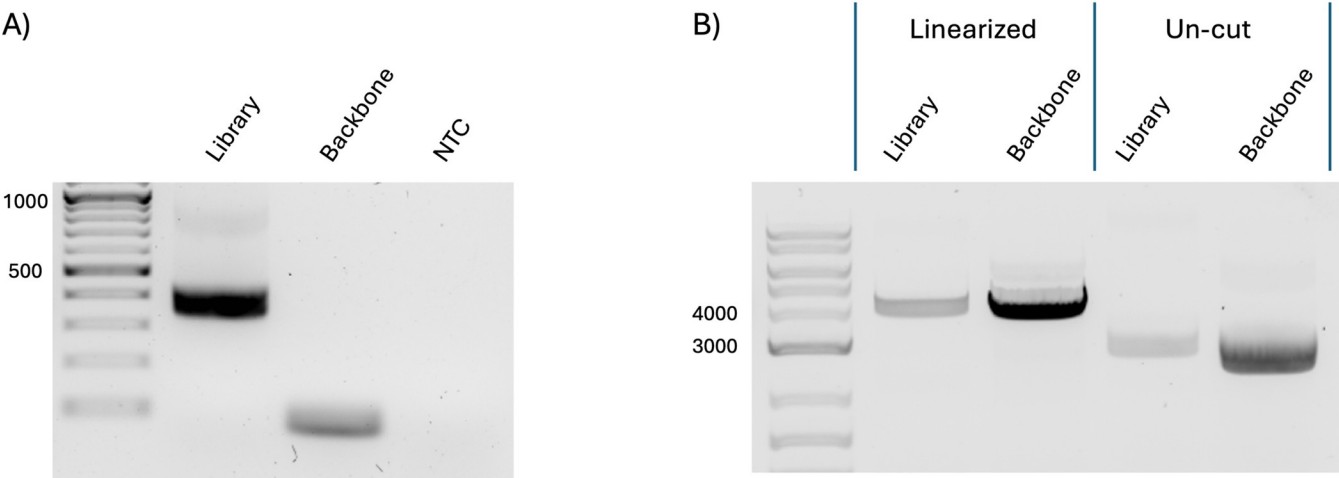

**Fig 3. Library fragment verification by PCR.** (A) A monobody library was created utilizing this method, and the length of the inserted sequence was confirmed using a primer that amplifies the insert. (B) Size verification by linearizing with Restriction Enzyme. The library DNA and backbone plasmid were digested to BamHI restriction enzyme to linearize the constructs for size verification by gel electrophoresis.

amino acid sequences in three solvent-exposed binding loops, akin to antibody CDR-H1/2/3, while conserving amino acids in the framework positions. This process aimed to enhance the efficiency and accuracy of library generation. The monobody library consisted of approximately 300 bases in length, and amplification was performed using primers that amplify a region of approximately 400 bases containing the monobody library. Amplification of an approximately 400 base insert was confirmed, and no backbone-derived amplicon (89bp) or any off-target products were detected (Fig 3A). This suggested that the presence of backbone and off-target products was below the detection limit of PCR. This absence of unwanted sequences ensures the purity and specificity of the library DNA. To further validate the efficiency of our library construction, we performed linearization using restriction enzymes. This step confirmed an increase in the overall size of the library DNA. The larger size indicated the monobody library DNA was indeed efficiently generated (Fig 3B). This optimized construction method confirmed the absence of unwanted sequences and validated the structural integrity and diversity of the monobody library, making it a robust tool for various biomedical applications.

## Discussion

Establishing a DNA library constitutes a critical procedure in biotechnology, genetic engineering, and basic research. This process involves extracting or synthesizing DNA fragments encoding for a collection of protein variants, then inserting these fragments into vectors. The DNA library is then introduced into host cells (often bacteria or yeast) to express the encoded protein variants, allowing for functional screens, enrichment, or selection of high performing protein variants.

SLiCE cloning has emerged as a versatile and effective method for molecular cloning, offering a more flexible alternative to traditional methods. Unlike conventional cloning techniques dependent on restriction enzymes, which require matching restriction sites and often involve laborious procedures–SLiCE cloning is not constrained by these requirements [28]. Traditional approaches often come with limitations like inefficient ligation and incomplete digestion, which pose challenges in constructing large DNA libraries and can lead to time-

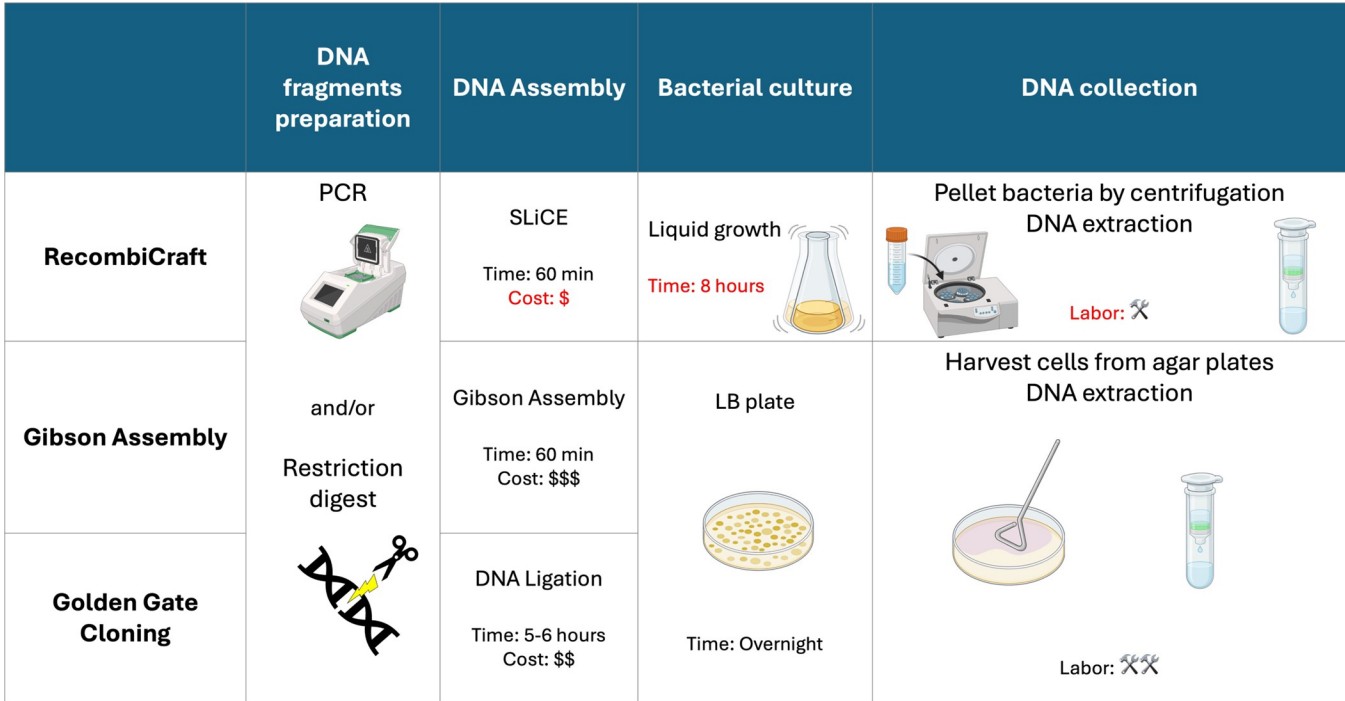

**Fig 4. Comparison of various library cloning methods.** A comparison of the steps, time, and costs involved in RecombiCraft and two other commonly used methods for constructing a DNA library.

consuming efforts [1]. Adopting liquid culture also reduces the cost and time required for library DNA extraction (Fig 4).

One significant advantage of SLiCE is its ability to accommodate homologous recombination even when extra sequences exist at the ends of backbone and insert (S1 Fig). This feature significantly enhances the flexibility of experimental designs, outperforming both traditional cloning techniques and modern methods, such as Gibson Assembly Unlike these methods, SLiCE cloning allows for the integration of DNA fragments into backbones without the need for precise end sequence matching or further modifications, thus improving adaptability and simplifying the cloning process [19, 22].

An essential element of SLiCE cloning is the homology length and polymerase selection involved in the process. Our results indicate that homologous regions of approximately 50 bases at both the 3' and 5' ends are efficient for library cloning, corroborated in studies by Zhang et al and Messerschmidt [19, 28]. While some exceptions have been reported, increasing the insert ratio generally correlates with higher cloning efficiency [19, 29].

The choice of polymerase plays a significant role in library cloning, as it can introduce bias in sequences amplified through PCR. Some polymerases falter when amplifying templates with high GC or AT content due to coverage bias. Previous studies demonstrated that the high-fidelity Kapa HiFi polymerase can amplify regions with AT-rich and GC-rich content, whereas other polymerases, like Phusion, may fail generally. However, Kapa HiFi polymerase does have a tendency to introduce errors in these challenging regions [30, 31]. Factors such as sequence composition, PCR reaction temperatures, and sequence length are also crucial considerations. The polymerase's origin may dictate its thermal stability during PCR [32, 33], and the accurate amplification of longer sequences may hinge on the enzyme's proofreading capability and processivity [34].

In recent years, it has been reported that liquid culture has superior library uniformity compared to plating or semi-solid culture [35], however, our experimental observations also pinpointed liquid culture as a potential source of bias in library generation. Variation in plasmid size, copy number, and specific sequence repeats can exert different metabolic demands on the host bacteria. Consequently, *E.coli* cells harboring certain plasmids may exhibit varied growth speed rates [36]. To address this concern, we assessed the growth bias attributable to incubation duration, revealing that the library uniformity could be compromised beyond 12 hours. Thus, we advocate for an 8-hour incubation period as the standard for this protocol. Adjusting incubation periods can be fine-tuned based on the specific goals of the experiment. Other research supports this, indicating that controlling growth bias is possible by carefully optimizing conditions like the composition of the growth medium and the levels of antibiotics used [24].

Verification of DNA libraries constructed via this method confirmed the successful insertion of only the intended library fragments and uniform linearized library assembly. While this example utilizes a monobody library, the technology is amenable to expansion for other library screening applications that involve cloning, including phage display. Additionally, the essential materials for this process are readily available in most molecular biology laboratories, offering a considerable cost advantage over commercial library construction kits.

In conclusion, our study introduces a novel DNA library construction methodology utilizing SLiCE, which we have named RecombiCraft. This approach facilitates the acquisition of sufficient DNA quantities for downstream applications by optimizing the liquid incubation period. When combined with existing techniques, this SLiCE-based method holds potential for a diverse range of applications.

## Supporting information

**S1 Table. The list of primers used in this study.**
(TIF)

**S1 Fig. The design flexibility of SLiCE cloning.**
(TIF)

**S1 Raw images.**
(PDF)

## Author Contributions

**Conceptualization:** Masako Harada.

**Data curation:** Yuki Kawai-Harada, Katherine Lauro.

**Formal analysis:** Yuki Kawai-Harada, Mehrsa Mardikoraem.

**Project administration:** Daniel Woldring, Masako Harada.

**Writing – original draft:** Yuki Kawai-Harada, Vasudha Nimmagadda, Quynh Tong, Kayla Bello.

**Writing – review & editing:** Yuki Kawai-Harada, Vasudha Nimmagadda, Daniel Woldring, Masako Harada.

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
