## [Decision Letter · Decision Letter 0]

29 Aug 2024

PONE-D-24-25871RecombiCraft Library construction: A novel method for DNA Library cloning and expansion using non-enzymatic single-step DNA recombination and liquid culturePLOS ONE

Dear Dr. Harada,

Thank you for submitting your manuscript to PLOS ONE. After careful consideration, we feel that it has merit but does not fully meet PLOS ONE’s publication criteria as it currently stands. Therefore, we invite you to submit a revised version of the manuscript that addresses the points raised during the review process.

We look forward to receiving your revised manuscript.

Kind regards,

Md Rajib Sharker, Ph.D.

Academic Editor

PLOS ONE

Journal Requirements:

2. Thank you for stating the following financial disclosure: This project received financial support from two sources: the Elsa U. Pardee Foundation, which focuses on cancer research and treatment funding, and the Michigan Translational Research and Commercialization (MTRAC) program, which provides early-stage funding to promote the commercialization of innovative technologies.

Additional Editor Comments (if provided):

The manuscript is well-organized and clearly written, presenting a study objective that is both relevant and valuable for publication. However, I strongly encourage the authors to address several critical issues within the manuscript. These concerns need to be thoroughly addressed to enhance the quality and rigor of the study. Therefore, I recommend that the manuscript undergo significant revisions before it can be considered for publication.

Reviewers' comments:

Reviewer's Responses to Questions

**Comments to the Author**

1. Is the manuscript technically sound, and do the data support the conclusions?

Reviewer #1: Yes

Reviewer #2: Yes

2. Has the statistical analysis been performed appropriately and rigorously? 

Reviewer #1: Yes

Reviewer #2: Yes

3. Have the authors made all data underlying the findings in their manuscript fully available?

Reviewer #1: Yes

Reviewer #2: Yes

4. Is the manuscript presented in an intelligible fashion and written in standard English?

Reviewer #1: Yes

Reviewer #2: Yes

5. Review Comments to the Author

Reviewer #1: In this study, the author presents a novel, rapid, and cost-effective method for constructing DNA libraries in E. coli. RecombiCraft combines seamless ligation cloning extract (SLiCE) with liquid culture amplification to minimize sequence biases. The method employs high-fidelity polymerase for the PCR synthesis of library backbones and inserts, and utilizes SLiCE for the assembly of DNA fragments, which are then introduced into E. coli via electroporation. RecombiCraft proves to be an economical and efficient tool, maintaining library uniformity and significantly shortening the library generation period, thus making it a valuable asset for genetic research and biotechnological applications.

The manuscript is well-organized and written, and the study's objective is worthy of publication. However, the reviewer strongly encourages the authors to address the following issues. Major amendments are required before the manuscript can be published. Constructive comments for revision are as follows:

Major concern

1. The author introduces RecombiCraft as an "innovative, rapid, and cost-efficient method" but doesn't provide enough specific details on what makes it unique compared to existing methods. Adding data or results could strengthen the claim.

Query 2: The manuscript would be significantly strengthened if you will do comet assay to check the DNA integrity after exposure and recovery of crude oil.

Query 3: The quality of some figures is not adequate for publication.

Query 4: In discussion section, rephrased some sentences for better readability and coherence.

Query 5: Made the comparison between SLiCE cloning and traditional methods clearer and more concise.

Minor comments:

Line 27: The phrase "significantly shorter library generation period" is vague. Providing specific time comparisons with other methods would make this claim more impactful.

Line 102: Library DNA" should either both be capitalized and not, depending on whether it's a proper noun. It might be better as "library DNA". Please check it throughout the manuscript.

Line 116-117: The sentence "To optimize library cloning, we created homologous sequences of varying lengths and compared their cloning efficiency." could be simplified or rephrased for clarity.

Line 149-151: Please omit correct the sentence in following ways.

The monobody library was designed based on a hydrophilic variant of human fibronectin, which improves biodistribution properties, making it well-suited for downstream applications in molecular imaging and therapeutics days post fertilization as you mentioned it before.

Line 152: The term "throughout" is omitted, which makes the sentence slightly redundant. Correct the sentence in following ways.

This hydrophilic monobody library included diverse amino acid sequences in three solvent-exposed binding loops, akin to antibody CDR-H1/2/3, while conserving amino acids in the framework positions.

Line 159: The phrase "at least backbone" is confusing and seems incorrect.

Line 164-165: he phrase "not only... but also" suggests two parallel ideas, but the way it's phrased here makes it slightly unclear. Correct the sentence in following ways.

This optimized construction method confirmed the absence of unwanted sequences and validated the structural integrity and diversity of the monobody library.

Line 172: Used "host cells" instead of "suitable cells" for clarity and consistency.

Line 192: Replaced "however" with a semicolon to improve readability and flow.

Line 198: Used "may" instead of "might" for a more formal tone and added "generally" to clarify that the correlation is not absolute.

Line 220-223: Changed "named RecombiCraft" to "which we have named RecombiCraft" for clarity. Changed "enables acquisition of sufficient DNA quantities" to "facilitates the acquisition of sufficient DNA quantities" for smoother readability. Changed "fine-tuning" to "optimizing" for a more formal tone and clearer meaning.

Reviewer #2: General comments of the manuscript:

The manuscript entitled, “RecombiCraft Library construction: A novel method for DNA Library cloning and expansion using non-enzymatic single-step DNA recombination and liquid culture”, is completely novel in biotechnology for sustainable development and wellbeing of the shorter library generation period. From the reviewer observation this research manuscript can be accepted with minor corrections in the esteemed journal, “PlosOne”.

The following correction is given below: The first figure requires high resolution since it seem a fuzzy like image to the reviewer.

1) Introduction: Accepted

2) Materials and methods: Accepted

3) Results: Accepted

4) Discussion: Accepted

5) Conclusion: Accepted

6. PLOS authors have the option to publish the peer review history of their article (what does this mean?). If published, this will include your full peer review and any attached files.

Reviewer #1: No

Reviewer #2: No

---

## [Author Response · Author response to Decision Letter 0]

30 Sep 2024

We appreciate the positive response from the editor/reviewer. The manuscript has been edited according to the suggestions. Point-by-point response is as follows:

Reviewer 1

Major concern

1. The author introduces RecombiCraft as an "innovative, rapid, and cost-efficient method" but doesn't provide enough specific details on what makes it unique compared to existing methods. Adding data or results could strengthen the claim.

>Thank you for your comment. We added the sentence to emphasize uniqueness in the abstract. Figure 4 also explains the difference between other traditional methods.

Query 2: The manuscript would be significantly strengthened if you will do comet assay to check the DNA integrity after exposure and recovery of crude oil.

>Thank you for your comment. This method proposes DNA library preparation, we think that figure 3 shows purity good enough for downstream applications.

Query 3: The quality of some figures is not adequate for publication.

>We have separately uploaded high-resolution images as figures.

Query 4: In discussion section, rephrased some sentences for better readability and coherence.

>We reassessed these sections and have modified the text to improve readability.

Query 5: Made the comparison between SLiCE cloning and traditional methods clearer and more concise.

>We have included additional text within the Discussion section to emphasize the superior aspects of this method.

Minor comments:

Line 27: The phrase "significantly shorter library generation period" is vague. Providing specific time comparisons with other methods would make this claim more impactful.

>We have now specified the minimum time to prepare DNA library in the Abstract.

Line 102: Library DNA" should either both be capitalized and not, depending on whether it's a proper noun. It might be better as "library DNA". Please check it throughout the manuscript.

> All spellings except the first word have been unified to "library."

Line 116-117: The sentence "To optimize library cloning, we created homologous sequences of varying lengths and compared their cloning efficiency." could be simplified or rephrased for clarity.

>This sentence has been rephrased: “We improved library cloning by creating different-length homologous sequences and comparing their cloning efficiency.“

Line 149-151: Please omit correct the sentence in following ways.

The monobody library was designed based on a hydrophilic variant of human fibronectin, which improves biodistribution properties, making it well-suited for downstream applications in molecular imaging and therapeutics days post fertilization as you mentioned it before.

> The statement has been corrected as indicated.

Line 152: The term "throughout" is omitted, which makes the sentence slightly redundant. Correct the sentence in following ways.

This hydrophilic monobody library included diverse amino acid sequences in three solvent-exposed binding loops, akin to antibody CDR-H1/2/3, while conserving amino acids in the framework positions.

> The statement has been corrected as indicated. 

Line 159: The phrase "at least backbone" is confusing and seems incorrect.

>”at least” was removed to correct this sentence.

Line 164-165: he phrase "not only... but also" suggests two parallel ideas, but the way it's phrased here makes it slightly unclear. Correct the sentence in following ways.

This optimized construction method confirmed the absence of unwanted sequences and validated the structural integrity and diversity of the monobody library.

> The statement has been corrected as indicated. 

Line 172: Used "host cells" instead of "suitable cells" for clarity and consistency.

> The statement has been corrected as indicated.

Line 192: Replaced "however" with a semicolon to improve readability and flow.

> The statement has been corrected as indicated.

Line 198: Used "may" instead of "might" for a more formal tone and added "generally" to clarify that the correlation is not absolute.

> The statement has been corrected as indicated.

Line 220-223: Changed "named RecombiCraft" to "which we have named RecombiCraft" for clarity. Changed "enables acquisition of sufficient DNA quantities" to "facilitates the acquisition of sufficient DNA quantities" for smoother readability. Changed "fine-tuning" to "optimizing" for a more formal tone and clearer meaning.

> The statement has been corrected as indicated.

Reviewer 2

Thank you very much for the positive comments.

We have uploaded a higher-resolution image tiff file.

---

## [Editor Report · Decision Letter 1]

2 Oct 2024

RecombiCraft library construction: A novel method for DNA library cloning and expansion using non-enzymatic single-step DNA recombination and liquid culture

PONE-D-24-25871R1

Dear Dr. Harada

We’re pleased to inform you that your manuscript has been judged scientifically suitable for publication and will be formally accepted for publication once it meets all outstanding technical requirements.

Kind regards,

Md Rajib Sharker, Ph.D.

Academic Editor

PLOS ONE

Additional Editor Comments (optional):

The authors did adequately address all the queries raised by Reviewer in their revised manuscript. The paper can now be considered for publication

---

## [Editor Report · Acceptance letter]

16 Oct 2024

PONE-D-24-25871R1 

PLOS ONE

Dear Dr. Harada, 

I'm pleased to inform you that your manuscript has been deemed suitable for publication in PLOS ONE. Congratulations! Your manuscript is now being handed over to our production team.

Kind regards, 

on behalf of

Dr. Md Rajib Sharker 

Academic Editor

PLOS ONE